# Elucidating the Link: Chronic Obstructive Pulmonary Disease and the Complex Interplay of Gastroesophageal Reflux Disease and Reflux-Related Complications

**DOI:** 10.3390/medicina59071270

**Published:** 2023-07-08

**Authors:** Xiaoliang Wang, Zachary Wright, Jiayan Wang, Stephen Roy, Ronnie Fass, Gengqing Song

**Affiliations:** 1Internal Medicine Residency Program, Joan C. Edwards School of Medicine, Marshall University, Huntington, WV 25701, USA; wangxi@marshall.edu (X.W.); wangji@marshall.edu (J.W.); roy31@marshall.edu (S.R.); 2Joan C. Edwards School of Medicine, Marshall University, Huntington, WV 25701, USA; wright476@marshall.edu; 3Department of Gastroenterology and Hepatology, Metrohealth Medical Center, Case Western Reserve University, Cleveland, OH 44106, USA; rfass@metrohealth.org

**Keywords:** COPD, environmental and occupational health and epidemiology, statistics, tobacco, emphysema

## Abstract

*Background and Objective:* Presenting chronic obstructive pulmonary disease (COPD) patients frequently report concurrent symptoms of gastroesophageal reflux disease (GERD). Few studies have shown a correlation between GERD and COPD. We aimed to examine the correlation between GERD and COPD as well as secondary related reflux complications, such as esophageal stricture, esophageal cancer, and Barrett’s esophagus. *Methods*: This population-based analysis included 7,159,694 patients. Patients diagnosed with GERD with and without COPD were compared to those without GERD. The enrollment of COPD included centrilobular and panlobular emphysema and chronic bronchitis. Risk factors of COPD or GERD were used for adjustment. Bivariate analyses were performed using the chi-squared test or Fisher exact test (2-tailed) for categorical variables as appropriate to assess the differences in the groups. *Results*: Our results showed that COPD patients had a significantly higher incidence of GERD compared to those without COPD (27.8% vs. 14.1%, *p* < 0.01). After adjustment of demographics and risk factors, COPD patients had a 1.407 times higher risk of developing non-erosive esophagitis (*p* < 0.01), 1.165 higher risk of erosive esophagitis (*p* < 0.01), 1.399 times higher risk of esophageal stricture (*p* < 0.01), 1.354 times higher risk of Barrett’s esophagus without dysplasia (*p* < 0.01), 1.327 times higher risk of Barrett’s esophagus with dysplasia, as well as 1.235 times higher risk of esophageal cancer than those without COPD. *Conclusions*: Based on the evidence from this study, there are sufficient data to provide convincing evidence of an association between COPD and GERD and its secondary reflux-related complications.

## 1. Introduction

The backflow of material from the stomach into the lower esophagus commonly causes complaints of indigestion or reflux and is the underlying mechanism behind gastroesophageal reflux disease (GERD). Clinically, GERD can be diagnosed based on atypical acid exposure in the esophagus on reflux monitoring testing or seen with endoscopy showing distinctive mucosal injury [1,2,3]. Symptoms not only include heartburn and the regurgitation of sour or bitter liquid but also non-burning chest discomfort that radiates towards the back, and dysphagia, or difficulty swallowing. A wide range of atypical symptoms can also occur, impacting the throat, larynx, and lungs. These include a sore throat, chronic cough, increased salivation, and shortness of breath [4,5,6]. The diversity of these symptoms can sometimes lead to diagnostic confusion, with GERD being mistaken for cardiovascular or pulmonary diseases due to symptom overlap. The prevalence of GERD in the United States is approximately 18–28% [7,8], resulting in an increased economic burden on the healthcare system and a significantly impaired quality of life [9]. GERD is a multifactorial condition influenced by various risk factors [3]. Commonly seen as one of the most prevalent risk factors is being overweight or obese [10]. Excess weight puts pressure on the abdomen, which can push the gastric contents retrogradely into the lower esophageal area. Additionally, abdominal fat may affect the functionality of the sphincter in the lower esophagus, further contributing to regurgitation. Another factor that plays a large role in the development of GERD is smoking [11]. Smoking diminishes lower esophageal sphincter (LES) function and reduces the production of saliva, which usually stabilizes the acidic stomach environment. The harmful chemicals in tobacco smoke can also irritate the esophagus and increase acid production, exacerbating reflux symptoms. Certain dietary choices and habits can also increase the risk of developing GERD [12]. Consuming a diet rich in unhealthy fats, citric acid from fruits, hot spices, caffeine, chocolate, fried foods, and beverages that are carbonated can relax the LES and trigger reflux. Eating large meals, especially close to bedtime, can also worsen symptoms by increasing pressure on the stomach and allowing acid to flow back into the esophagus. Hiatal hernia, an abnormality where a portion of the gastrointestinal tract, usually the stomach, protrudes into the chest through the diaphragm, is also considered a risk factor for GERD [13]. A hernia can disrupt the normal functioning of the LES, allowing stomach acid to reflux into the esophagus.

Chronic obstructive pulmonary disease (COPD) is a prevalent respiratory condition characterized by airflow obstruction and breathing difficulties [14,15,16]. According to the 2023 Global Initiative for Chronic obstructive lung disease (GOLD), COPD is defined as a diverse respiratory condition marked by chronic symptoms (such as dyspnea, cough, expectoration, and/or exacerbations) resulting from abnormalities in the airways such as bronchitis or bronchiolitis, and/or emphysema. These abnormalities lead to persistent airflow obstruction, often exhibiting a progressive nature [16]. The definite diagnosis of COPD is mainly based on pulmonary function tests (PFT) (spirometry), with a forced expiratory volume in 1 s (FEV1)/forced expiratory volume (FVC) ratio less than 70% being the general criteria [16]. The health status, disease progression, and prognosis of individuals with COPD are significantly affected by exacerbations. According to the new definition by GOLD 2023, which embraced the recent consensus proposal known as the Rome proposal [17], ECOPD is characterized as an event where dyspnea and/or cough and sputum worsen over a period of up to 14 days. This event may also involve symptoms such as tachypnea and/or tachycardia and is often associated with increased local and systemic inflammation resulting from airway infection, pollution, or other insults to the airways [16].

Several risk factors contribute to the development and progression of COPD. One of the primary risk factors is cigarette smoking [18], with approximately 85% of COPD cases having a history of smoking [19]. Prolonged exposure to tobacco smoke damages the lungs and leads to inflammation and narrowing of the airways, ultimately causing airflow limitation [20]. Environmental factors also play a crucial role. Occupational exposure to harmful substances like dust, chemicals, and fumes can contribute to the development of COPD [21]. Additionally, indoor and outdoor air pollution, including biomass fuel used for cooking and heating, can further exacerbate the condition [22]. Genetics and alpha-1 antitrypsin deficiency are other risk factors [23]. A deficiency in this protein can lead to early-onset COPD, even in individuals who have never smoked. Moreover, frequent respiratory infections, such as repeated bouts of pneumonia, especially *Pseudomonas aeruginosa*, which is significantly associated with worsening FEV1 [24], can increase the risk of developing COPD [25]. Beyond its primary impact on the lungs, COPD is often associated with various comorbidities which significantly affect the overall health and prognosis of individuals with the disease [26]. One of the most common comorbidities seen in COPD patients is cardiovascular disease, such as coronary artery disease, hypertension, pulmonary artery hypertension, and heart failure [27]. These conditions share risk factors with COPD, such as smoking, systemic inflammation, and oxidative stress. Furthermore, COPD is frequently associated with metabolic disorders like diabetes mellitus and osteoporosis [28]. The systemic inflammation present in COPD can lead to insulin resistance, contributing to the development of diabetes. Similarly, the chronic inflammatory state and corticosteroid use in COPD increases the risk of osteoporosis, resulting in an elevated susceptibility to fractures [29]. Additionally, mental health disorders, such as anxiety and depression, are prevalent comorbidities in COPD patients [30]. The progressive nature of COPD, accompanied by symptoms like dyspnea and functional limitations, can lead to psychological distress, social isolation, and reduced quality of life, thereby worsening mental health conditions [31]. According to the CDC, the age-adjusted prevalence of COPD remained stable from 2011 to 2020, ranging from 5.9% to 6.6% [32]. Despite this, the age-adjusted mortality of COPD patients has decreased consistently since 1999, with a mortality rate of 166.2 per 10,000 patients recorded in 2019 [32]. This improvement is attributed to advanced treatment methods and the development of new medications [16,33].

Heartburn and regurgitation are common symptoms in patients with COPD, with individuals who experience heartburn also reporting chronic coughing and dyspnea [34,35,36]. Due to the high prevalence of GERD and COPD, the potential interaction between the two conditions has been the subject of extensive investigation [35,37,38]. Several studies have shown that GERD is an independent risk factor for COPD exacerbations [35]. In a prospective study involving 82 patients with COPD exacerbation, acid reflux was significantly correlated to a six times greater risk of developing an exacerbation of COPD [36]. This conclusion was also supported by a systematic review and meta-analysis conducted in Brazil, which found that individuals who had GERD had a relative risk of 7.6 for COPD exacerbations [38]. In another meta-analysis study, treatment of GERD with a proton-pump inhibitor (PPI) was associated with reduced frequency of COPD exacerbation and related mortality [37].

Several studies have demonstrated an increased prevalence of GERD in patients with COPD [39,40,41,42]. A cross-sectional study conducted in Korea found that 28% of patients with COPD had GERD and that most COPD medications were associated with acid reflux [41]. Another study of 151 patients with and without COPD found that a significantly higher proportion of COPD patients reported symptoms of acid reflux and dysphagia than those without COPD [42]. Additionally, medications such as theophylline were found to increase the risk of GERD symptoms in patients with COPD [40]. However, limited data determined there to be a distinct connection between COPD and developing GERD or GERD-related complications.

This retrospective study, utilizing information from a large national database, was designed to evaluate if individuals with COPD had a greater risk of developing GERD. Secondarily, we intended to analyze the relationship concerning COPD and complications of GERD, including Barrett’s esophagus without or with dysplasia, esophageal cancer, and esophageal stricture.

## 2. Materials and Methods

### 2.1. Database

The National Inpatient Sample (NIS) is a comprehensive nationwide healthcare database that encompasses data from almost 7 million hospital stays each year. This publicly accessible database includes information from various payers and is not weighted. Established by the healthcare and utilization project (HCUP), it was devised to approximate inpatient cost, utilization, outcomes, and quality. The database estimates a 20% stratified sample of all community hospital discharges in the United States, not including long-term acute care hospitals and rehabilitation. A retrospective analysis was performed using this database.

### 2.2. Data Collection and Outcomes

This study included a total of 7,159,694 adult patients who were admitted to the hospital. The patients were categorized into different groups based on their diagnosis of GERD (ICD-10 K21.9 and K21.0) and the presence or absence of chronic bronchitis, emphysema, or other chronic COPD (ICD-10 J40, J41, J42, J43, J44). The comparison was made between patients with GERD and those without GERD. Within the GERD group, subtypes such as non-erosive reflux disease (NERD) (ICD-10-CM 21.9) and erosive esophagitis (EE) (ICD-10 K21.0) were identified. Complications related to GERD, such as esophageal stricture, Barrett’s esophagus with or without dysplasia, and esophageal cancer, were only considered in patients with a diagnosis of GERD. Patients with asthma or bronchiectasis were excluded from the control groups. The diagnosis of COPD was determined based on pulmonary function tests. Subjects with a history of foregut surgeries, uncontrolled type 2 diabetes (T2DM), eosinophilic esophagitis, and infective esophagitis were also excluded. Variable adjustment analysis was conducted using risk factors for COPD (e.g., cigarette smoking history) and risk factors for GERD (e.g., hiatal hernia, cigarette smoking, and obesity). Demographic data, including age, race, gender, obesity, and smoking history, were collected. The odds ratio of GERD or GERD-related complications in patients with COPD was evaluated, with GERD diagnosis or its subtypes (NERD or EE) as the dependent factor and COPD diagnosis as the variable factor. Complications of GERD, such as esophageal stricture, Barrett’s esophagus (with or without dysplasia), or esophageal cancer, were analyzed as dependent factors compared to the variable factors. All diagnoses included or excluded in this study were determined using the ICD-10 code.

### 2.3. Statistical Analysis

All demographic and risk factor data obtained from the NIS for this study were categorical, thus presented as various cases and percentages. The association between GERD and COPD was analyzed using chi-squared analysis, while the association between GERD complications in individuals with and without COPD was investigated. To assess the risk of GERD and GERD complications in patients with COPD, multivariate logistic regression analysis was performed, presenting odds ratios. Covariates such as age, gender, race, cigarette smoking, hiatal hernia, and obesity were adjusted to minimize the influence of confounding factors. A 2-sample test for equal proportions was conducted, with a significance level of *p* < 0.05 considered as statistically significant. The statistical analysis was carried out using IBM SPSS 28.0.1.1.

## 3. Results

This study included a total of 7,159,694 hospitalized patients, among whom 1,179,759 individuals were diagnosed with GERD, with and without COPD (283,416 with COPD and 896,305 without COPD) (Figure 1). It was observed that GERD patients with COPD were generally older compared to those without COPD, with mean ages of 69.4 ± 0.1 and 62.6 ± 0.1, respectively (*p* < 0.05). No significant differences were found in terms of obesity and hiatal hernia between GERD patients with and without COPD. However, the GERD with COPD group had a significantly higher proportion of cigarette smokers (25.3% vs. 10.9%, *p* < 0.01) and oxygen-dependent patients (17.5% vs. 1.5%, *p* < 0.01) compared to the GERD without COPD group (Table 1).

Patients with COPD exhibited a significantly higher likelihood of having NERD compared to those without COPD (OR 1.407, 95% CI 1.399–1.414, *p* < 0.001). The incidence of NERD was 27.7% among patients with COPD and 14.1% among those without COPD, indicating a statistically significant difference (*p* < 0.001). In terms of EE, the incidence was 101.7 per 10,000 patients in the COPD group and 53.6 per 10,000 patients in the non-COPD group. Patients with COPD had a significantly higher risk of EE (OR: 1.165, 95% CI: 1.133–1.199, *p* < 0.001) (Table 2 and Figure 2).

In relation to GERD-related complications, it was found that GERD patients with COPD had a significantly elevated risk of developing esophageal stricture (OR 1.399, 95% CI 1.321–1.481, *p* < 0.001). The incidence of esophageal stricture was 24.7 per 10,000 patients in GERD patients with COPD, whereas it was 9.6 per 100,000 in those without COPD, indicating a statistically significant difference (*p* < 0.01) (Table 2 and Figure 2). Moreover, GERD patients with COPD demonstrated a higher likelihood of developing Barrett’s esophagus without dysplasia compared to those without COPD (OR 1.354, 95% CI: 1.312–1.397, *p* < 0.001). The incidence of Barrett’s esophagus without dysplasia was 84.6 per 10,000 in GERD patients with COPD, whereas it was 33.4 per 10,000 in those without COPD (*p* < 0.01). Furthermore, the incidence of Barrett’s esophagus with dysplasia was 1.8 per 10,000 in GERD patients with COPD and 0.8 per 10,000 in those without COPD (*p* < 0.01). GERD patients with COPD were at a significantly higher risk of Barrett’s esophagus with dysplasia (OR: 1.235, 95% CI: 1.002–1.521, *p* < 0.05) (Table 2 and Figure 2).

Additionally, it was observed that GERD patients with COPD faced a markedly increased risk of esophageal cancer (OR 1.327, 95% CI 1.204–1.462, *p* < 0.01). The incidence of esophageal cancer was 8.4 per 10,000 among GERD patients with COPD, whereas it was 3.5 per 100,000 in those without COPD, representing a statistically significant difference (*p* < 0.01) (Table 2 and Figure 2).

Furthermore, upon further analysis, it was revealed that the incidence of GERD or its complications was significantly higher in both female and male patients with COPD (Appendix A). Additionally, COPD patients across various age ranges demonstrated a significantly increased incidence of GERD or GERD complications. However, no significant difference in the incidence of Barrett’s esophagus with dysplasia was found among COPD patients aged over 55 years (Appendix A).

## 4. Discussion

From our findings, one of the most important outcomes was the greater incidence and risk of GERD, and its associated complications in patients with COPD, when matched to individuals without COPD. The information from this groundbreaking study is the first to utilize a large inpatient database to examine the association between GERD and COPD patients and provides clear evidence of a correlation. The result is found to have a more substantial meaning when you consider the adjustment of several significant confounding variables, such as hiatal hernia, race, gender, smoking, age, and obesity. These results are consistent with previous smaller studies, such as one that found a GERD prevalence of 37% in patients with COPD when compared to the control group of 18% [43]. Another study (16 healthy volunteers and 42 males with COPD) found a three-times greater GERD prevalence in individuals with COPD when compared to the control group [39]. However, a study using esophageal pH monitoring and upper endoscopy by Kempainen et al. found that although 57% of patients with COPD had GERD, only higher BMI was predictive of GERD, with an odds ratio of 1.2 [44]. Another study (1486 patients) analyzing PFT suggested that only inspiratory capacity, not FEV1, was associated with a greater probability of GERD in individuals with COPD [45]. Our study is the first to demonstrate that COPD can be an independent risk factor for developing GERD.

Importantly, we also found that COPD is associated with a higher risk of GERD related complications. Patients with COPD were 40% more likely to develop esophageal stricture, 35% more probable to acquire Barrett’s without dysplasia, 23% more likely to develop Barrett’s esophagus with dysplasia, and 32% more likely to progresses to esophageal cancer when compared to those without COPD. This highlights the importance of considering COPD as an independent risk factor for GERD complications. Even though we were unable to unveil the exact underlying physiological mechanism behind COPD and GERD secondary complications, there was an obvious connection among them. Although no previous studies have shown the exact prevalence of esophageal stricture among patients with COPD, it has been reported that patients with COPD exacerbation are more likely to have dysphagia [46]. Individuals with even mild COPD were found to have dysphagia symptoms [47]. A case-control study by Menon et al. found a significantly increased risk of Barrett’s esophagus among patients with COPD [48]. They found that the hazard ratio of Barrett’s esophagus in patients with COPD was 1.58 on multivariable analysis. However, no significant association between COPD and the risk of developing esophageal cancer was identified in their study [48].

The underlying mechanisms linking COPD to an increased risk of GERD and its complications remain unclear. However, some potential explanations have been proposed, such as alterations in esophageal physiology, changes in diaphragm anatomy in COPD patients, and molecular and signaling changes related to COPD.

The relationship between the anatomy of the diaphragm and the esophagus plays a role in the development of GERD and its complications in individuals with COPD. The distal esophagus passes through the diaphragm at a junction called the esophageal hiatus, and diaphragmatic contraction during inspiration functions as an external sphincter that compresses this junction, increasing lower esophageal sphincter pressure and acts as an anti-reflux mechanism [24,25]. In contrast, when the diaphragm is relaxed, the lower esophageal sphincter has transient relaxation, increasing the potential for GERD. COPD can lead to contractile protein degradation and muscle fiber remodeling, which can impair the contractile function and endurance of the diaphragm. This decrease in contractile force and elasticity can result in a loss of the diaphragm’s function as an external sphincter, leading to impaired anti-reflux mechanisms and, ultimately, GERD [26,27].

It has been suggested that the increased respiratory effort due to severe hyperinflation in COPD could lead to elevated abdominal and trans diaphragmatic pressure [49], potentially disrupting transient relaxations of the lower esophageal sphincter (LES) and leading to acid reflux [50]. A study conducted by Mittal et al. on 12 healthy subjects found that partial expiration, which mimics the breathing pattern in COPD patients, disrupted esophageal peristalsis, prolonged esophageal transit time, and decreased esophageal clearance [51].

The underlying mechanisms behind the higher risk of GERD and its associated complications in patients with COPD are yet to be fully understood. However, research has shed some light on the molecular basis of GERD, with findings pointing towards a role in inflammation cytokines and oxidative stress. Studies have shown that these factors are involved in the development of GERD [52,53]. In patients with COPD, inflammation cytokines and oxidative stress have also been implicated in its pathophysiology [54,55]. It is believed that cigarette smoking triggered the inflammation cascade in patients with COPD, eventually becoming self-amplified as more inflammatory cells are recruited [56]. Several sources of reactive oxygen species (ROS) are generated in patients with COPD, including exogenous sources mainly from smoking and endogenous sources produced by inflammatory cells [57]. The accumulation of oxidative stress in the setting of COPD may contribute to developing GERD.

This study has several important clinical implications for healthcare providers in managing patients with COPD. Firstly, increased awareness of GERD in COPD; the study highlights the need for healthcare providers to be vigilant about the presence of GERD symptoms in COPD patients. Recognizing the high incidence of GERD in COPD can help clinicians in making an accurate diagnosis and appropriate management decisions. Patients presenting with symptoms such as heartburn, regurgitation, or chest discomfort should be evaluated for GERD, especially if they have a known diagnosis of COPD. Secondly, multiple studies have shown that addressing GERD symptoms in COPD patients may have a positive impact on respiratory outcomes. GERD can exacerbate COPD symptoms, leading to increased coughing, wheezing, and dyspnea. By identifying and managing GERD in these patients, healthcare providers may potentially alleviate respiratory symptoms and improve overall lung function, resulting in better disease control and quality of life. Furthermore, this study’s findings emphasize the importance of regular surveillance for reflux-related complications in COPD patients with GERD. Healthcare providers should consider screening these patients for conditions such as esophageal stricture, Barrett’s esophagus, and esophageal cancer. Early detection of these complications allows for timely intervention and management, potentially improving patient outcomes and prognoses.

Our study has several limitations that should be acknowledged. Firstly, while the NIS, a nationwide inpatient database, provided comprehensive data, it did not include outpatient information, which constitutes a significant portion of clinical practice. Secondly, the diagnosis of GERD and its associated complications relied on ICD-10 codes documented in various hospital systems and electronic medical records. It is assumed that the diagnosis of each GERD complication, such as esophageal stricture, Barrett’s esophagus, and esophageal cancer, was supported by imaging, endoscopy, and pathology findings. Additionally, the risk factors for GERD, including smoking, obesity, and hiatal hernia, were identified using ICD-10 codes; however, the timeframe of these risk factors could not be determined. Lastly, the selection of subjects with different stages of chronic bronchitis and emphysema, assumed to be diagnosed based on pulmonary function tests (PFT), was also based on ICD-10 codes.

To further validate and expand on the findings of the current study, we are planning to conduct a retrospective study within our local hospital. This retrospective study aims to confirm the association between COPD and GERD, as well as to explore additional factors that may contribute to the development of reflux-related complications. By accessing the electronic medical records of patients, we will gather comprehensive data, including past medical history, current medications, and patient outcomes. In addition to the retrospective study, we also have plans to conduct a prospective study involving two major hospitals in our local area. This prospective study will involve a collaboration between gastroenterologists and pulmonologists with the objective of screening COPD patients presenting with moderate to severe GERD symptoms. The primary aim of this study is to determine the true prevalence of GERD in COPD patients and investigate the correlation between COPD and reflux-related complications. By performing screening esophagogastroduodenoscopy (EGD), we will be able to identify and assess the presence of esophageal complications such as esophagitis, stricture, Barrett’s esophagus, or even early esophageal cancer. The combined approach of the retrospective and prospective study will provide a comprehensive understanding of the relationship between COPD and GERD, including the prevalence of GERD in COPD patients and the impact of GERD on the development of esophageal complications.

## 5. Conclusions

In conclusion, this population-based analysis provides evidence of a significant correlation between gastroesophageal reflux disease (GERD) and chronic obstructive pulmonary disease (COPD), as well as several secondary related reflux complications. The study included a large cohort of 7,159,694 patients, allowing for robust statistical analysis. The results demonstrate that COPD patients have a significantly higher incidence of GERD compared to those without COPD, indicating a potential link between these two conditions. Furthermore, after adjusting for demographics and risk factors, COPD patients were found to have a higher risk of developing various reflux complications. Specifically, COPD patients had a higher risk of non-erosive esophagitis, erosive esophagitis, esophageal stricture, Barrett’s esophagus without dysplasia, Barrett’s esophagus with dysplasia, and esophageal cancer when compared to those without COPD. These findings suggest that the presence of COPD may contribute to the development or progression of GERD and its associated complications.

The findings underscore the significance of recognizing GERD as a comorbidity in individuals with COPD and emphasize the need for increased awareness regarding the potential consequences of GERD-related complications on their overall well-being. Healthcare professionals should be vigilant in monitoring and managing GERD symptoms in COPD patients, as early intervention and treatment may help mitigate the risk of developing severe reflux-related complications. Further research is warranted to elucidate the underlying mechanisms and potential causative factors linking COPD and GERD. A better understanding of this relationship could lead to the development of targeted interventions and management strategies that improve the quality of life and clinical outcomes for patients with both conditions.

## Figures and Tables

**Figure 1 medicina-59-01270-f001:**
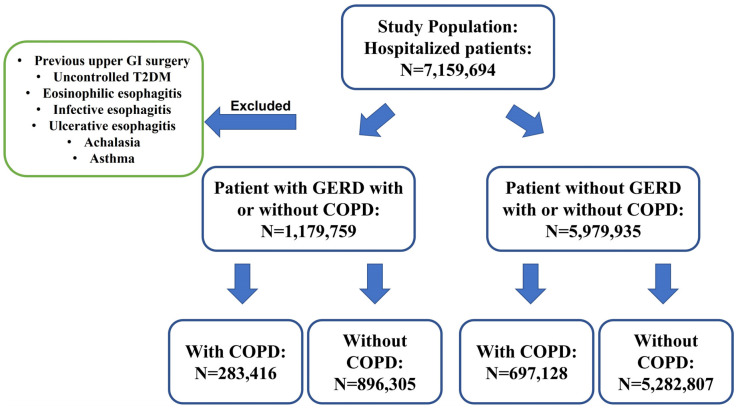
Sample selection and data design flowchart.

**Figure 2 medicina-59-01270-f002:**
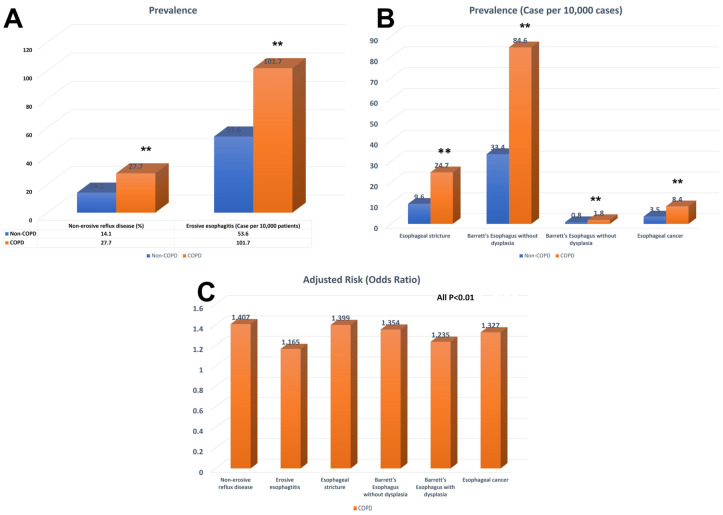
Bar graph of prevalence and odds ratio for COPD patients with GERD or GERD-related complications. (**A**) Prevalence of NERD, EE in patients with or without COPD. (**B**) Prevalence of esophageal stricture, Barrett’s esophagus and esophageal cancer in patients with or without COPD. (**C**) Adjusted odds ratio of NERD, EE, esophagus stricture, Barrett’s esophagus and esophagus cancer in patients with different stages of COPD. NERD, Non-erosive reflux disease; EE, Erosive esophagitis; COPD, chronic obstructive pulmonary disease. Adjusted for age, sex, race, obesity, hiatal hernia, and history of smoking.

**Table 1 medicina-59-01270-t001:** Demographics between GERD with and without COPD. GERD, gastroesophageal reflux disease; COPD, chronic obstructive pulmonary disease; w/, with; w/o, without.

	GERD w/ COPD	GERD w/o COPD	*p* Value
**Age**	69.4 ± 0.1	62.6 ± 0.1	<0.05
**Sex**			
**Female**	163,422 (57.7%)	526,068 (58.7%)	<0.01
**Male**	119,994 (42.3%)	370,237 (41.3%)	<0.01
**Race**			
**White**	223,968 (79.0%)	641,169 (71.5%)	<0.01
**Black**	31,758 (11.2%)	112,906 (12.6%)	<0.01
**Hispanic**	11,897 (4.2%)	69,712 (7.8%)	<0.01
**Asian**	2402 (0.8%)	15,969 (1.8%)	<0.01
**Risk factors**			
**Home oxygen**	49,713 (17.5%)	13,145 (1.5%)	<0.01
**Obesity**	56,610 (20.0%)	181,703 (20.3%)	>0.05
**Smoking**	71,730 (25.3%)	98,072 (10.9%)	<0.01
**Hiatal hernia**	14,340 (5.1%)	50,383 (5.6%)	>0.05

**Table 2 medicina-59-01270-t002:** The prevalence and odds ratio for GERD and GERD-associated complications in patients with COPD. OR, odds ratio; GERD, gastroesophageal reflux disease; COPD, chronic obstructive pulmonary disease; CI, confidence interval. Adjusted for age, sex, race, hiatal hernia, obesity, and smoking.

	Case	Prevalence (%) or Case per 10,000 Patients	Adjusted OR	95% CI	*p* Value
	**Nonerosive reflux disease**				
**COPD**	276,281	27.70%	1.407	1.399–1.414	<0.01
**Non-COPD**	868,762	14.10%			
	**Erosive esophagitis**				
**COPD**	7348	101.7	1.165	1.133–1.199	<0.01
**Non-COPD**	28,361	53.6			
	**Esophageal stricture**				
**COPD**	1782	24.7	1.399	1.321–1.481	<0.01
**Non-COPD**	5100	9.6			
	**Barrett’s esophagus w/o dysplasia**				
**COPD**	5081	84.6	1.354	1.312–1.397	<0.01
**Non-COPD**	17,625	33.4			
	**Barrett’s esophagus w/ dysplasia**				
**COPD**	128	1.8	1.235	1.002–1.521	<0.05
**Non-COPD**	404	0.8			
	**Esophageal cancer**				
**COPD**	605	8.4	1.327	1.204–1.462	<0.01
**Non-COPD**	1852	3.5			

## Data Availability

The data presented in this study are available on request from the corresponding author. The data are not publicly available due to patient and hospital information privacy and the requirement of H.CUP.

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
