# Peer review of "Elucidating the Link: Chronic Obstructive Pulmonary Disease and the Complex Interplay of Gastroesophageal Reflux Disease and Reflux-Related Complications"

_medicina, 2023, doi:10.3390/medicina59071270_

Round 1
Reviewer 1 Report
Dear
1. There are several grammatical errors. For example, you should change "correlation between GERD and COPD" to "a correlation between GERD and COPD". "Risk factors of COPD or GERD was used for adjustment. " to "Risk factors of COPD or GERD were used for adjustment.", etc.
2. Criteria for cases are incomplete.
3. Please change Figure 2 to a new one with white background.
4. Please add the prevalence based on age, sex, ...
5. Please re-edit table 2 without duplicates for "COPD case Case per 10,000 patients Adjusted OR 95% CI p value".
6. Please add the full name of each abbreviation below each figure or table.
7. What is the clinical significance?
8. What is a future comment?
9. Please update the number of references to recent 5-year citations.
10. You could add the association of GERD with comorbidities and medication utilization in patients with COPD?
There are several grammatical errors. For example, you should change "correlation between GERD and COPD" to "a correlation between GERD and COPD". "Risk factors of COPD or GERD was used for adjustment. " to "Risk factors of COPD or GERD were used for adjustment.", etc.
Author Response
- There are several grammatical errors. For example, you should change "correlation between GERD and COPD" to "a correlation between GERD and COPD". "Risk factors of COPD or GERD was used for adjustment. " to "Risk factors of COPD or GERD were used for adjustment.", etc.
-They have been corrected. Thank you.
- Criteria for cases are incomplete.
-We have amended the 'Data Collection and Outcome' subsection within the Methods section to incorporate more precise criteria for participant selection.
- Please change Figure 2 to a new one with white background.
-It has been changed accordingly.
- Please add the prevalence based on age, sex, ...
- They have been included in the supplemental material Table 1 and Figure 1.
- Please re-edit table 2 without duplicates for "COPD case Case per 10,000 patients Adjusted OR 95% CI p value".
-It has been revised accordingly.
- Please add the full name of each abbreviation below each figure or table.
-They have been added.
- What is the clinical significance?
- We added the clinical significance of this study in the discussion part.
- What is a future comment?
- We have incorporated into the final section of our discussion an outline of future research directions.".
- Please update the number of references to recent 5-year citations.
- We have made significant revisions to the references as suggested by the reviewer. However, we have opted to maintain several pivotal citations to preserve the integrity of the original author's work.
- You could add the association of GERD with comorbidities and medication utilization in patients with COPD?
- We value your input. Acknowledging the correlation between comorbidities like T2DM and obesity, and GERD, these factors have been considered in our analysis of the GERD-COPD relationship. Our current study primarily addresses GERD and its complications, while comorbidities such as depression, anxiety, hypertension, and osteoporosis will be the focus of our upcoming retrospective study conducted in local hospitals. A significant limitation of our current study is the lack of medication usage data in our database, which we aim to overcome in the upcoming study by accessing comprehensive patient information at our local hospital.
Reviewer 2 Report
I have the following comments
1. It is an overestimation to mention the complications in the title and the text without caution; despite significant p values, the RR is very close to 1.
2. The exacerbation term is mentioned twice in the introduction but its description/definition is not precise.
3. Line 53-58: the CDC reference and access date??
4. Line 59 and 60: what is the reference?
5. More references are needed like GOLD 2023
Author Response
- It is an overestimation to mention the complications in the title and the text without caution; despite significant p values, the RR is very close to 1.
We agree with the reviewer's observation that the risk of COPD patients developing specific GERD complications, such as Barrett's esophagus with dysplasia, is only 23% compared to the control group, with a statistically significant p-value of <0.05. It's critical to highlight that the clinical implications of these specific complications are relatively less significant compared to other associated risks. As such, we have revised the title to "Elucidating the Link: Chronic Obstructive Pulmonary Disease and the Complex Interplay of Gastroesophageal Reflux Disease and Reflux-related Complications."
- The exacerbation term is mentioned twice in the introduction but its description/definition is not precise.
-Thank you for the comments. We have revised the introduction part with a clearer definition of COPDE based on the GOLD 2023.
- Line 53-58: the CDC reference and access date??
-Reference added, this is the website link: https://www.cdc.gov/copd/data-and-statistics/national-trends.html
- Line 59 and 60: what is the reference?
Our apologies for the previously missing citations. We have now included references 33 to 35 in the appropriate sections.
- More references are needed like GOLD 2023
-Thank you for the comments. More updated references such as GOLD 2023 have been added.